

# *Mastigodiaptomus galapagoensis* n. sp. (Crustacea: Copepoda: Diaptomidae), a possibly extinct copepod from a crater lake of the Galápagos archipelago

Manuel Elías-Gutiérrez[1], Miriam Steinitz-Kannan[2], Eduardo Suárez-Morales[1] and Carlos López[3]

[1] Aquatic Ecology and Systematics, El Colegio de la Frontera Sur, Chetumal, Quintana Roo, Mexico
[2] Department of Biological Sciences, Northern Kentucky University, Highland Heights, KY, USA
[3] Escuela Superior Politécnica del Litoral (ESPOL), Centro de Agua y Desarrollo Sustentable, Guayaquil, Ecuador

Corresponding author
Eduardo Suárez-Morales,
esuarez@ecosur.mx

## ABSTRACT

**Background**. A new species of a Neotropical diaptomid copepod is described based on individuals recovered from a small, almost forgotten collection of unique plankton samples from El Junco, a crater lake in San Cristóbal island, Galápagos archipelago. This copepod was regularly reported (1966–2004) as an abundant zooplankter in the lake, but it was not found in subsequent plankton surveys (2007–2018), and its specific identity remained unknown. In 2020, it was declared extinct because of introduced fish predation, rotenone treatment, and other major disturbances. The taxonomic examination of these invaluable specimens allowed us to recognize them as representing an undescribed species of the freshwater diaptomid genus *Mastigodiaptomus Light, 1939*.

**Methods**. Here, we describe the new species from El Junco crater lake, located in the San Cristóbal island a part of the Galápagos archipelago, collected with plankton nets. The description is based on detailed morphology, based on SEM and light microscopy.

**Results**. The taxonomic examination of these invaluable specimens allowed us to recognize them as representing an undescribed species of the freshwater diaptomid genus *Mastigodiaptomus Light, 1939*. The new species was readily assigned to this genus and is distinguished from its known congeners by details of (1) the male right fifth leg terminal claw and aculeus, (2) spiniform processes pattern of the right geniculate antennule segments 10–16, (3) length and structure of the spiniform process of the antepenultimate segment of the male right antennule, and (4) details of the dorsal process on the female fourth pediger. This finding represents the first report of this Neotropical copepod genus outside its original biogeographic region, the third species of a diaptomid copepod reported from insular freshwater systems, the southernmost record of *Mastigodiaptomus*, and the only freshwater calanoid in the Galápagos. The intriguing presence of this chiefly Neotropical copepod genus here could be related either to (1) human agency linked to pirate activities, commercial travelling by Spaniard ships, whaling activities, and intense tortoise hunting in San Cristóbal island. In the past, El Junco was the only freshwater source 600 nautical miles around, or (2) zoochory of resistant dormant stages passively transported by more than 65 migrating bird species known to settle in San Cristóbal. These two hypotheses cannot be properly tested at

this time, so the explanation of the presence of this copepod will remain as a new open question in the fascinating natural history of the Galápagos.

## INTRODUCTION

Tucked away in the drawer of a cabinet holding a research collection at Northern Kentucky University (NKU) are vials containing preserved large red copepods from the Galápagos Islands. All the samples are from a single lake, El Junco, in San Cristóbal Island, the only permanent freshwater lake in the Galápagos archipelago. The copepods were collected in separate expeditions between 1966 and 2004, complementing paleoclimatic studies of the archipelago (*Bush et al., 2010*). Plankton collections were made with a 63 μm net in 1966 by P. Colinvaux and by M. Steinitz-Kannan in 1977, 1978, 1984, 1988, 1991, 1996, and 2004. They were preserved in Lugol's iodine and examined primarily for diatoms and other algae, yet the predominance of the large copepods in all the samples was hard to ignore. The first mention of the copepod was by *Steinitz-Kannan (1979)*, who noted its resemblance with *Notodiaptomus amazonicus* and observed that in July 1966, July 1977, and December 1978, there was a disproportionate large number of adults (up to 80% of the sampled population) compared to nauplii larvae and copepodites, presumably resulting from a slow turnover of the population. According to *Corgosinho et al. (2019)*, this copepod population is the only calanoid species in inland waters of the archipelago and needs taxonomic verification. Recently, Dr. Miriam Steinitz-Kannan brought the existence of these copepods to the attention of Dr. Carlos López who examined the specimens from the 2004 collection and recruited Dr. Manuel Elías-Gutiérrez and Dr. Eduardo Suárez-Morales to aid with the taxonomic identification.

All the specimens examined and described here were collected in September 2004. In 2005, tilapia was introduced in this lake, and it appears to have caused the extinction of this copepod (see *López et al., 2021*). The diaptomid copepod genus *Mastigodiaptomus* is the most diverse in the northern Neotropical region; it currently includes about 15 nominal species (*Gutiérrez-Aguirre et al., 2020*). The genus was originally described by *Light (1939)*, who designated *M. albuquerquensis* (Herrick, 1895) as the type species. In different taxonomic accounts of the Neotropical freshwater copepods, this nominal species was deemed as the most widespread and diverse diaptomid from the southern Nearctic region to the northern Neotropics (*Reid, 1990*; *Suárez-Morales et al., 1996*; *Suárez-Morales & Reid, 1998*). Recent works (*Gutiérrez-Aguirre & Cervantes-Martínez, 2016*; *Gutiérrez-Aguirre et al., 2020*) have recognized that *M. albuquerquensis* and probably *M. nesus* likely contain different, yet undescribed, cryptic species in Middle America corresponding with the northern limit of the Neotropics.

Most species of *Mastigodiaptomus* are known from inland freshwater habitats. Only one species of this genus has been recorded from insular freshwater systems: *M. nesus*. Another

species, *Mastigodiaptomus ha*, Cervantes-Martínez, 2020 (in *Gutiérrez-Aguirre et al., 2020*) was also found in a lagoon of Cozumel Island, in the Yucatan Peninsula (*Gutiérrez-Aguirre et al., 2020*).

Here, we describe a new species of *Mastigodiaptomus* and compare it with its known congeneric species. The new species is here described following the current morphological descriptive standards for diaptomid copepods (*Gutiérrez-Aguirre & Cervantes-Martínez, 2013*; *Mercado-Salas et al., 2018*). Describing a new species that is now deemed extinct from the iconic Galápagos Islands was possible only because specimens were preserved in a small university research collection. Such collections are in danger of disappearing (see *Jardine, 2013*), together with the 1966 Galápagos samples collected by Colinvaux. Except for the material rescued by one of us (MSK) and taken to NKU, all these collections were discarded because of disinterested institutional decisions. In addition to describing a new copepod species from the Galápagos Islands, this paper, therefore, draws attention to the importance of preserving old university research collections.

## MATERIALS & METHODS

Plankton samples were obtained from El Junco, a small crater lake located in San Cristóbal Island, in the Galápagos archipelago (0°53′42.80″S; 89°28′47.34″W) on September 2, 2004, under permission granted by the Galápagos National Park as part of Project PC 08-04. Samples were collected using a small plankton net with a 63 μm mesh size and fixed with Lugol. The specimens examined were deposited in the Collection of Zooplankton (ECO-CHZ) held at El Colegio de la Frontera Sur (ECOSUR), Chetumal, Mexico, where they are available for further examination. A subsample from the same collection with numerous specimens is also deposited in the same collection (ECO-CHZ). The rest of the collections and original samples are at NKU.

The specimens were dissected and drawn in a BX51 microscope with a camera lucida attached. All observations were made under normal illumination, Nomarski and phase contrast techniques. Other six specimens were dried with increasing concentrations (50, 60, 70, 80 90, and 100%) of HDMS (hexamethyldisilazane) and covered with gold. Observations of these individuals were made in a JEOL 6010 scanning electron microscope (SEM) in the Chetumal Unit of ECOSUR.

All drawings and SEM photographs are included for the description of this species. The lsid registered in ZooBank for this publication is: urn:lsid:zoobank.org:pub:B1999A13-FD8B-4750-988B-5BE4C7167AA9.

### Nomenclatural acts

The electronic version of this article in Portable Document Format will represent a published work according to the International Commission on Zoological Nomenclature (ICZN), and hence the new names contained in the electronic version are effectively published under that Code from the electronic edition alone. This published work and the nomenclatural acts it contains have been registered in ZooBank, the online registration system for the ICZN. The ZooBank Life Science Identifiers (LSIDs) can be resolved and the associated information viewed through any standard web browser by appending the

LSID to the prefix http://zoobank.org/. The LSID for the new species nomenclatural act is registered in ZooBank lsid as: urn:lsid:zoobank.org:act:91C915F7-E18F-489D-9D06-DBC0E4201C7D

The online version of this work is archived and available from the following digital repositories: PeerJ, PubMed Central and CLOCKSS.

## RESULTS

### Systematics

Class Copepoda Milne Edwards, 1840
Order Calanoida Sars, 1903
Family Diaptomidae Baird, 1850
Genus *Mastigodiaptomus Light, 1939*
***Mastigodiaptomus galapagoensis* sp. nov.**

urn:lsid:zoobank.org:pub:B1999A13-FD8B-4750-988B-5BE4C7167AA9

**Material examined.** Adult male holotype, specimen undissected, vial with 1 ml of propylene-glycol, plankton sample, El Junco crater lake, San Cristóbal island, Galápagos archipelago (0°53′42.80″S; 89°28′47.34″ W), collected September 2, 2004 by M. Steinitz-Kannan (ECO-CH-Z 11820). Allotype, Adult female, undissected, same locality, date and collector, vial with propylene-glycol (ECO-CHZ-11821). Paratypes: One male specimen, dissected, mounted on semi-permanent slide sealed with DePex mounting medium (Dibutyl phthalate or Gurr) (ECO-CH-Z-11822), five males in vial with propylene-glycol (ECO-CH-Z-11822). One female dissected, mounted in semi-permanent slide sealed with DePex mounting medium (Dibutyl- phthalate or Gurr) (ECO-CH-Z-11823), six adult females in vial with propylene-glycol (ECO-CH-Z-11823). One male and one female, gold-coated, mounted on SEM stub(ECO-CH-Z-11824).

**Diagnosis.** Medium-sized *Mastigodiaptomus* with 24-segmented female antennules, female fourth pediger with large blunt process on dorsal surface. Male geniculate right antennule 22-segmented, antepenultimate segment with long, straight acute process reaching beyond distal margin of succeeding segment 21. Male right antennule middle section with large spinous processes on segments 13 and 14, process on segment 11 longer than that on segment 10. Segment 16 lacking process. Female fifth leg with inner spiniform process of second exopodal segment with spinulate inner margin, outer margin smooth; endopod 2-segmented, reaching halfway of first exopodal segment. Right male fifth leg with long, slender aculeus about half as long as elongate distal claw. Right endopod as long as first exopodal segment.

### Description

**Female** (Figs. 1B; 2A–2M; 3A–3J) with total body length 1,328 μm (range 1,310–1,360 μm), from anterior end of cephalothorax to posterior end of anal somite, fourth pediger with large, conical dorsal process inserted off-central axis (Figs. 1B; 3A), pointing out to

right side. Urosome 3-segmented, including proximally expanded genital-double somite, short preanal and cylindrical anal somites (Figs. 1B; 2M). Rostrum with short, stout beak-like rostral points lacking distal filaments (Fig. 3C). Fifth pediger posterolateral wings moderately asymmetrical, left longer than right, distalmost with long spine, proximalmost with small spinule (Figs. 1B, 3A, 2I); right wing with two spines (Figs. 1B, 3J). Pedigerous somites 4–5 dorsally fused, incomplete suture on lateral margins; conspicuous blunt process arising semi-laterally from fourth pediger; process tilted to right side of body (Figs. 1B, 3A, 3I). Genital-double somite with single acute spine on right proximal 1/3, left margin with slightly smaller spine, directed backwards (Figs. 2M, 3J). Genital-double somite ventrally expanded at genital field; distal operculum plate well-developed, with symmetrical rounded flaps, proximal opercular plate reduced (Fig. 3J). Caudal rami about as long as preceding anal somite. Rami rectangular, about twice as long as broad, inner and outer margins smooth, lightly pilose. Rami symmetrical, armed with 5 caudal setae, dorsal seta reduced, slender, about 0.5 times as long as ramus (Fig. 2M). All egg-carrying females with one or two eggs.

**Antennule** 24-segmented, reaching anterior margin of anal somite. Segmental armature (s = seta, ae = aesthetasc) as: 1(1s, 1ae), 2(2s, 1ae), 3 (2s, 1ae), 4-5(1s), 6 (1s, 1 ae), 7 (1s), 8 (2s), 9(1s), 10 (1s), 11 (2s), 12-13 (1s), 14 (1s, 1ae), 15(1s), 16(1s, 1ae), 17-18 (1s), 19 (1s, 1ae), 20-21 (1s), 22(2s), 23 (2s), 24(2s), 25(4s + 1ae) (Figs. 2A, 3B).

**Antenna** with subquadrate coxa armed with single seta. Basis rectangular, carrying 2 subequally long setae on distal inner margin. Endopod 2-segmented, first segment cylindrical, armed with two subequal setae at distal 1/3; second endopod segment bilobed, proximal lobe armed with 9 setae, distal lobe with 6. Exopodal ramus 8-segmented, second expressed segment with pseudosegments; setal formula as: 1, 2, 1, 1, 1, 1,1, 4. (Figs. 2B; 3F). Last segment with three apical and one proximal setae.

**Mandible** with gnathobase bearing 7 short, rounded teeth, innermost margin with short, lightly spinulose dorsal seta, outermost margin weakly expanded (Fig. 2C). Palp biramous. Coxa with the palp. Endopod 2-segmented; first endopodal segment subquadrate, armed with 6 setae, second segment armed with 5 apical setae apically and one proximal (Fig. 2D). Exopod 4-segmented, setal formula; 1, 1, 1, 3

**Maxillule** with precoxal arthrite armed with 14 spiniform setae, coxal epipodite with 10 setae, two proximalmost elements short, curved; coxal endite subquadrate, armed with four apical setae. Basis with basal endite carrying 4 setae, inner lobe with 4 setae and basal exite represented by single seta. Endopod 2-segmented, each segment carrying four subequally long setae. Exopod 1-segmented, armed with six subequal setae (Fig. 2E).

**Maxilla** with two precoxal endites armed with 3 and 2 setae, respectively. Two succeeding coxal lobes each with 3 apical setae lacking spinules at insertion. Basis with well-developed allobasis, bearing 3 slender setae. Endopod 4-segmented, first segment with single seta, second segment reduced, with 3 apical setae (Fig. 2F).

**Maxilliped** with precoxa and coxa fused, with four lobes, setal formula: 1, 2, 3, 4 . Coxal distal rounded process ornamented with a row of minute spinules. Basis bearing 3 slender setae, distalmost being longest; endopod 5-segmented, setal formula: 2, 3, 2, 2, 5 (Fig. 2G).

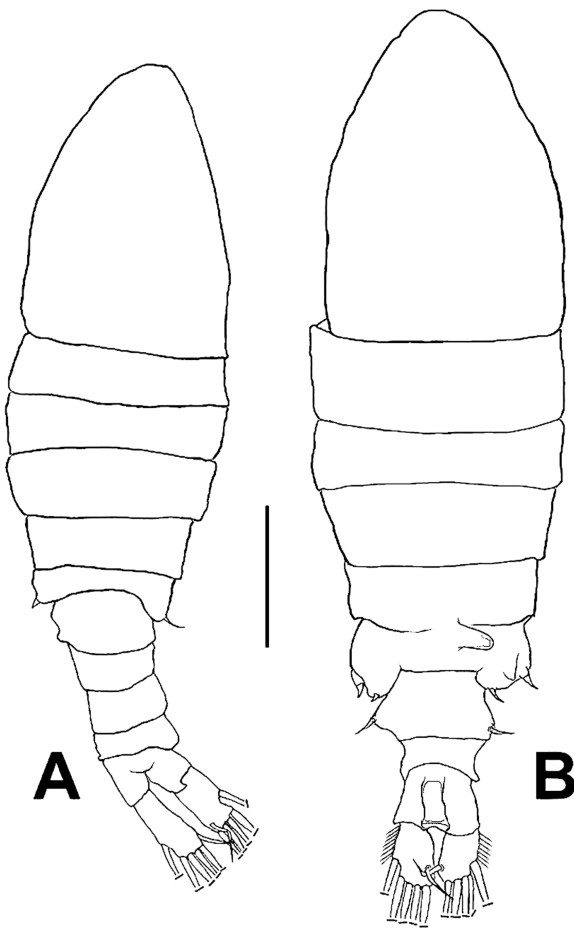

**Figure 1** *Mastigodiaptomus galapagoensis.* **n. sp., habitus.** (A) Male holotype. (B) Female allotype. Scale bar = 100 μm.

**P1** (Fig. 2H), endopod 2-segmented, exopod 3-segmented. Coxa with relatively long, plumose inner coxal seta reaching beyond distal margin of basis. Basis unarmed. Outer margins of second and third exopodal segments pilose.

**P2** (Fig. 2I), exopod and endopod 3 -segmented, endopod relatively shorter, reaching slightly beyond distal margin of second exopodal segment. Coxal seta, long, reaching beyond proximal end of first endopodal segment. Outer exopodal spines short, pinnate.

**P3** (Fig. 2J), coxal seta as in P2; basis unarmed. Endopodal and exopodal rami 3-segmented, Endopod shorter than exopod, reaching about halfway of third exopodal segment. Outer spine on exopodal segments robust, pinnate.

**P4** (Fig. 2K), coxal seta as in P2 and P3; basis unarmed. Endopodal and exopodal rami 3-segmented, Endopod shorter than exopod, reaching about proximal 1/3 of third exopodal segment. Outer spine on exopodal segments as in P3.

Fig. Female P5 (Figs. 2L; 3D, 3E, 3G, 3H): Symmetrical. Coxa robust, subquadrate, distal margins expanded; coxa armed with strong seta on outer margin; seta reaching distal margin of basis. Basis short, unarmed. Endopod 2- segmented (Fig. 3D), reaching halfway
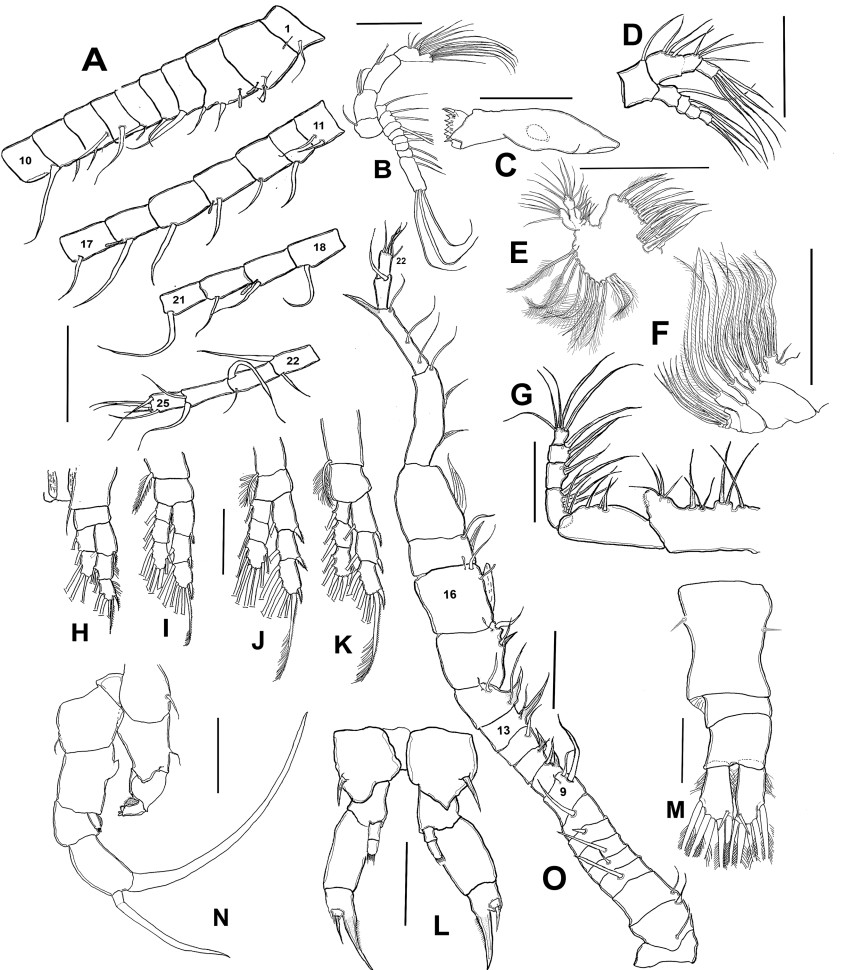

**Figure 2** *Mastigodiaptomus galapagoensis* n. sp., adult female and male from El Junco lake, Galapagos Islands. (A) Antennule sections 1 –10, 11–17, 18–21, and 22–25. (B) Antenna. (C) Mandble gnathobase. (D) Mandible palp. (E) Maxillule. (F) Maxilla. (G) Maxilliped. (H) Leg 1. (I) Leg 2. (J) Leg 3. (K) Leg 4. (L) Female fifth leg. (M) Female urosome, dorsal view. (N) Male fifth leg, anterior view. (O) Male right antennule. Scale bars = 100 µm.

of first exopodal segment, with apical brush-like row of short, slender elements. Exopod 3-segmented, first segment robust, rectangular, unarmed. Second segment with strong inner claw ornamented with spinules along inner margin only (Figs. 2L, 3G). As usual in genus, third segment carrying one short and one long setae (Figs. 3G, 3H).

**Male** (Figs. 1A, 2N, 2O, 4A–4K) smaller than female, total body length 1,234 µm (range 1,200–1,270 µm excluding caudal rami (Fig. 1A). Body slender, complete suture between pedigers 4–5 (Fig. 1A). Rostrum with simple, acute rostral processes (Fig. 4C). Left thoracic wing weakly projected, bearing acute spine (Figs. 1A; 4E). Right thoracic wing slightly projected, with single long spine (Fig. 1A). Caudal rami with 6 caudal setae, including reduced dorsal seta (Fig. 1A). Right A1 22-segmented, armature per segment (s = seta, ae = aesthetasc, spp = spiniform process, ms = modified seta) as: 1(1s,

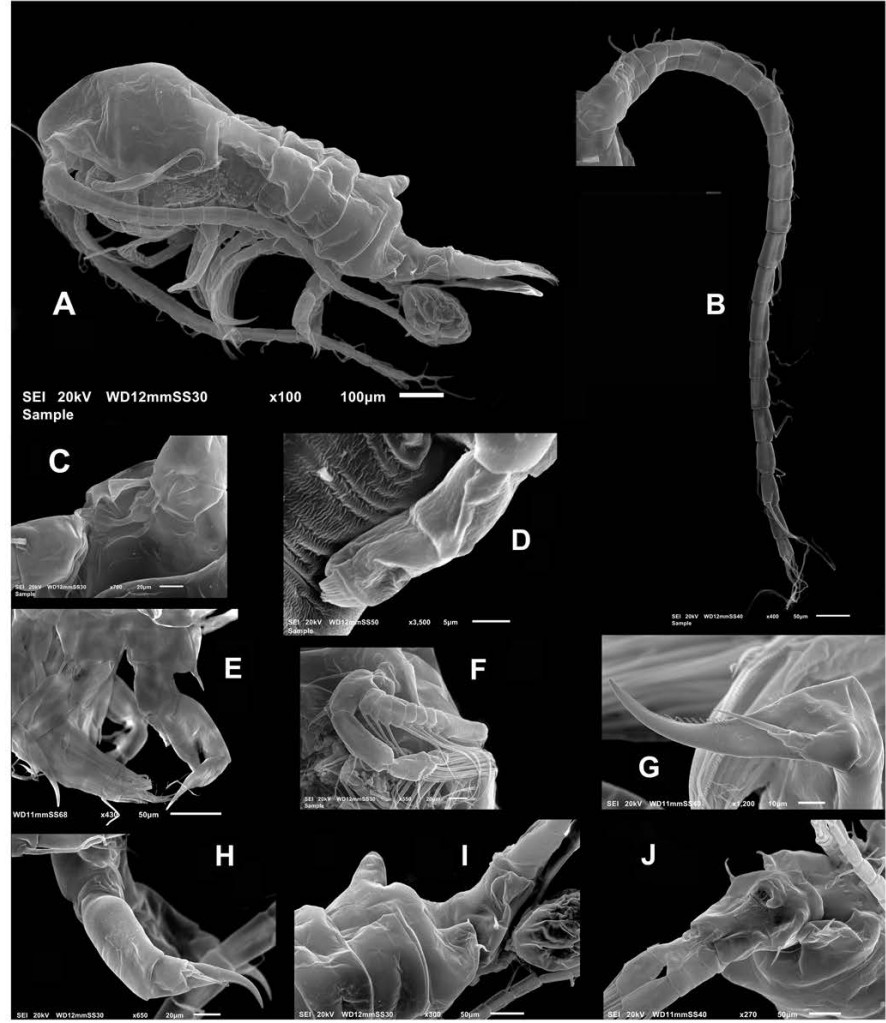

**Figure 3** *Mastigodiaptomus galapagoensis.* **n. sp., adult female, SEM- prepared individual from El Junco lake, Galapagos Islands.** (A) Habitus, lateral view. (B) Complete antennule. (C) Rostrum, ventral view. (D) Endopod, fifth leg. (E) Fifth leg showing coxal spiniform setal element. (F) Antenna. (G) Fifth leg exopod. (H) Same, lateral view. (I) Tilted dorsal conical process on fourth pedigerous somite, lateral view. (J) Genital double-somite, ventral view.

1ae), 2(2s), 3–5(1s), 6(1s), 7(1s), 8(1s), 9 (2s+1ae), 10(1s+1spp), 11(1s +spp), 12(1s +1ae), 13(1s+1spp+1ae), 14(1ae+2s+1sp), 15(2s,1ae,1spp), 16(2s+1ae+1spp), 17(1s, 1sp), 18(1ms), 19(1s+1ms), 20(2s+1spp), 21(2s), 22(5s). Segment 20 (antepenultimate) with long, beak-like distal process, reaching distal margin of succeeding segment 21 (Figs. 2O, 4B). Left A1, mouthparts, and P1–P4 as in female.

**Male right P5** (Figs. 2N, 4F–4K), coxa with short, slender outer seta. Basis inner margin with low, triangular process at proximal half. Endopod short, unsegmented, barely reaching distal end of first exopodal segment, armed with brush-like apical group of elements (Figs. 2N, 4G). First exopodal segment short, subquadrate. Second exopodal segment rectangular,
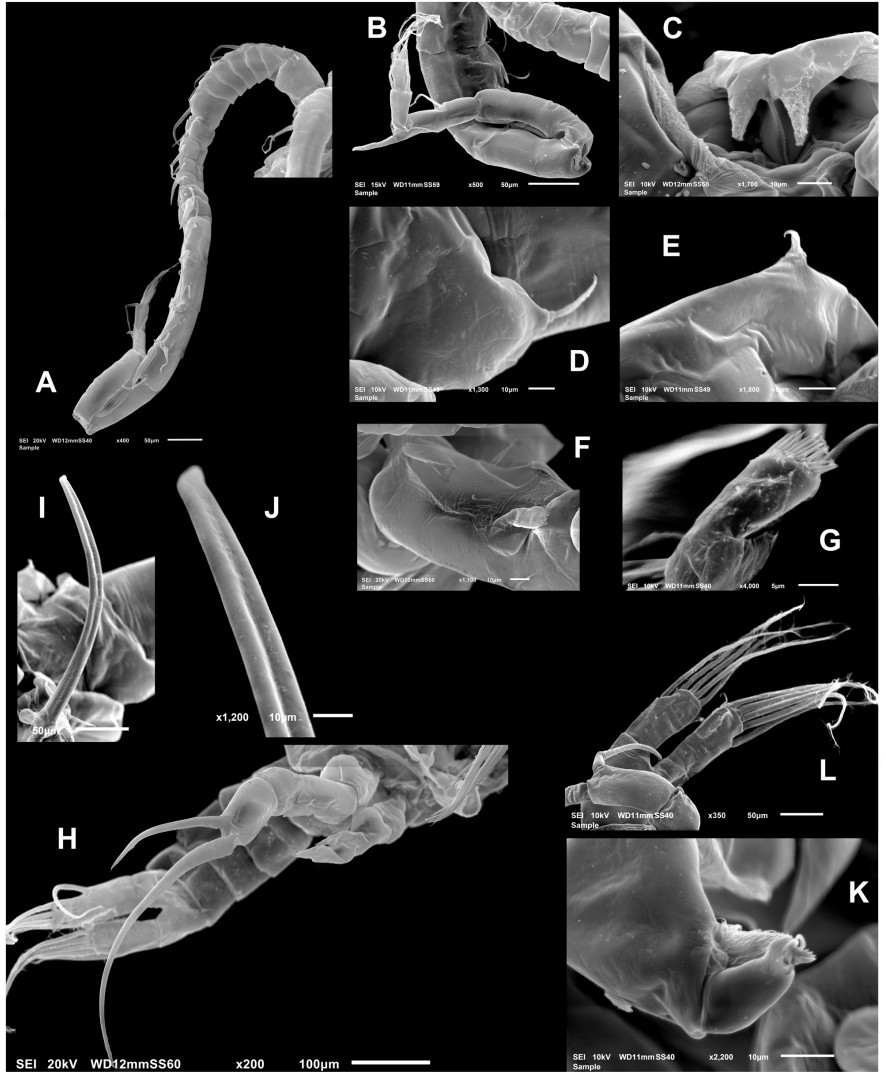

**Figure 4** *Mastigodiaptomus galapagoensis.* **n. sp., adult male, SEM-prepared individual from El Junco lake, Galapagos Islands.** (A) Right antennule. (B) Same, detail of distal segments and spinous process on segment 20. (C) Rostral points, ventral view. (D) Curved spine on right wing of fifth pediger. (E) Spine on left wing of fifth pedigerous somite. (F) Endopod of right fifth leg. (G) Endopod of left fifth leg. (H) Fifth leg, general view. (I) Detail of claw groove. (J) Tip of claw, right fifth leg (K) Detail of male left leg.

1.8 times as long as broad; terminal claw very long and slender, about 1.7 times as long as ramus, with smooth margins and distal 2/3 with shallow longitudinal groove (Figs. 2N, 4H–4J). Aculeus long, slender, curved, about twice as long as bearing segment; aculeus inserted subapically, with smooth margins (Figs. 2N, 4H).

**Male Left P5**, coxa with short outer seta inserted at distal half. Basis robust, rectangular, with distal triangular expansion covering part of first exopodal segment and armed with short basis seta. Endopod as long as first exopodal segment (Fig. 2N). Second exopodal segment with pilose inner pad-like process, segment tapering distally into curved spiniform

process ending with fine denticles in one side and adjacent terminal crown-like process with 4–5 short denticles (Fig. 4K).

## DISCUSSION

### Remarks on genus assignation

The new species was assigned as a member of the genus *Mastigodiaptomus* because its possession of the characters defined by *Light (1939)*, *Wilson (1959)*, and *Dussart & Defaye (1995)* for this genus, including: (1) distal segment of male left fifth leg shorter than proximal segment, (2) proximal pad of same ramus pilose, (3) second segment of male fifth leg right exopod short, not twice as long as wide, (4) terminal claw longer than exopodal ramus including basis, claw strongly bent or curved, (5) female and left male segment 11 and left male antennule with two setae, (6) coxa of female P5 with long, spiniform seta, (7) endopod of right male P5 as long as or slightly longer than right first exopodal segment, (8) right male antennule with spiniform process on segments 10, 11 and 13–16, (9) both male fifth leg endopods short, with distal ends bearing row of slender setules, (10) right male antennule with antepenultimate segment carrying stout curved process, major spine on segment 13, those on segments 10 and 11 smaller; spinous process on segment 14 arising proximally on segment, (11) third segment of fifth female exopod small but distinct, bearing long, robust spine-like seta and one short, vestigial; endopods short, truncate, usually 2-segmented.

### Comparative remarks on the new species

Currently, we know 15 species of *Mastigodiaptomus*, all of them described from the north Neotropics and south Nearctics. Following the recent keys of *Gutiérrez-Aguirre et al. (2020)* and *Suárez-Morales et al. (2020)* for the known species of *Mastigodiaptomus*, the new species keys down to either *M. nesus* (*Bowman, 1986*), *M. texensis* (*Wilson, 1953*) or *M. alexei* Elías-Gutiérrez, 2020 (in *Gutiérrez-Aguirre et al., 2020*) but has greater resemblance with the former in the left fifth exopod (compare Fig. 4K from here and Fig. 4G from *Gutiérrez-Aguirre et al., 2020*). Our species from Galápagos can be distinguished from these congeners by the length of the spinous process on the antepenultimate segment of the right male antennule; in the new species it is as long as the carrying segment whereas in *M. nesus*, the process is about half the length of the segment (*Bowman, 1986*, Figs. 4I, 4G; *Cervantes-Martínez et al., 2005*, Fig. 2H); also, the dorsal process on pediger 4 is anteriorly concave in *M. nesus* (*Bowman, 1986*) *vs.* not concave in the new species. *Mastigodiaptomus galapagoensis* differs from *M. texensis* in several respects. In *M. texensis* the right male P5 endopod is long, reaching beyond the distal margin of first exopodal article, *vs.* a shorter endopod in the new species, barely reaching the distal margin of the first exopod. In *M. texensis* the female P5 endopod reaches well beyond midlength of succeeding first exopodal segment (*Suárez-Morales et al., 2020*, Fig. 21.11H), whereas the same structure is clearly shorter in *M. galapagoensis*, barely reaching the proximal 1/3 of the first exopodal segment (Fig. 2L). Overall, the new species combines a unique set of characters including: (1) spinous process on antepenultimate segment of the right male antennule very long, longer than carrying segment (Fig. 4B), (2) conspicuous conical

process on female pediger 4 arising from off medial axis and tilted to the right side of the body, (3) remarkably long aculeus of right male fifth leg, almost as long as exopodal ramus (Figs. 2N, 4H). Only a few congeners share with the new species a dorsal process on the fourth pedigerous somite; according to *Gutiérrez-Aguirre & Cervantes-Martínez (2013)* and *Gutiérrez-Aguirre, Cervantes-Martínez & Elías-Gutiérrez (2014)*, this character has been recorded in populations of *M. amatitlanensis* (Wilson, 1941); *M. texensis*, *M. albuquerquensis* (see *Bowman, 1986*, Figs. 3K, 3M; *Gutiérrez-Aguirre, Cervantes-Martínez & Elías-Gutiérrez, 2014*, Fig. 5F), *M. ha* Cervantes-Martínez 2020 (in *Gutiérrez-Aguirre et al., 2020*), *M. patzcuarensis* (*Gutiérrez-Aguirre, Cervantes-Martínez & Elías-Gutiérrez, 2014*) (*Gutiérrez-Aguirre, Cervantes-Martínez & Elías-Gutiérrez, 2014*, Figs. 7G, 7H), and *M. montezumae* (see *Santos-Silva, Elías-Gutiérrez & Silva-Briano, 1996*). Only in the latter two the dorsal process is tilted, but in *M. patzcuarensis* the process on the antepenultimate segment of the male right antennule is about half the length of the carrying segment; the same process is even shorter in *M. montezumae* (*Santos-Silva, Elías-Gutiérrez & Silva-Briano, 1996*, Fig. 25), thus contrasting with the clearly longer process found in the new species. This process is known to be constant in the same species (*Gutiérrez-Aguirre et al., 2020*). Some additional affinities of the new species with its known congeners include a long aculeus, as long as or longer than bearing segment; this character is present in *M. siankaanensis* (see *Mercado-Salas et al., 2018*, Fig. 9E), *M. cihuatlan* Gutiérrez-Aguirre, 2020 (see *Gutiérrez-Aguirre et al., 2020*, Fig. 11F), in a single population of *M. albuquerquensis* (see *Bowman, 1986*, Fig. 4M), and in *M. nesus* (see *Bowman, 1986*, Figs. 4J, 4K; *Gutiérrez-Aguirre et al., 2020*, Fig. 2E). The Galápagos *Mastigodiaptomus* is a yet undescribed species that can be readily distinguished from its congeners. Unfortunately, the preservation method used at that time did not allow performing DNA analyses of the specimens.

## Ecology

El Junco crater lake, at 675 m elevation in San Cristóbal island, is the only permanent freshwater natural lake in the Galápagos archipelago. It is a small system 270 m in diameter, and a surface area of 6 ha (*Lopez et al., 2018a*). The maximum depth is 6 m, and depth can drop to 3 m during severe droughts. There is a small channel that drains the water during strong El Niño events, associated with heavy rains. The water temperature oscillates from 18.5 °C (in the cool season, July to December) to 25 °C in the warm season (January to June) (*Obando, 2009*). The lake is polymictic due to the strong winds at the crater. It is oligotrophic with its water chemistry like rainwater. Its Secchi disk transparency is low (about 2 m) due to humic coloration. The pH is low but varies depending on the time of day because of photosynthesis since the water is poorly buffered. In 2004 the pH values varied between 5.77 and 8.09. The phytoplankton is dominated by desmids (mostly *Hyalotheca* and *Staurastrum),* and an endemic species of diatom, *Frustulia galapagosaxonica*, most abundant during high water periods (*Conroy et al., 2009*).

Other zooplankton species found in El Junco, in the same sample (September 2, 2004) were a few individuals of the *Moina micrura* group, not previously reported (*López, Steinitz-Kannan & Segers, 2018b*). Other species reported from the same place are *Simocephalus exspinosus*, *Ceriodaphnia* sp., *Chydorus* sp., *Alona* sp., and *Euryalona orientalis* (reported

first by *Obando, 2009*). *Keratella cochlearis* was an abundant rotifer in the lake (*López, Steinitz-Kannan & Segers, 2018b*).

Adjacent areas of the lake were an important tortoise hunting site. Tortoises were finally extirpated from the site, and in the early 20th century, the lake became an important freshwater source (*Bush et al., 2022*). It is known that whaler expeditions regularly visited the islands since 1790; by the late 1890's, introduced livestock was widespread on the island. In the early 2000s the Galápagos National Park put fences to exclude the cattle from the area surrounding the lake, and started the reintroduction of endemic flora, in particular *Miconia robinsoniana* and removal of exotic species from the lake's basin. (*Bush, Restrepo & Collins, 2014*; *Bush et al., 2022*).

## Biogeographical notes

Tropical diaptomids were probably present in the Caribbean Plate alignment and subsequent integration of Central America (*Iturralde-Vinent & MacPhee, 1999*; *Braszus et al., 2021*) (30–70 Mya). The origin of this Neotropical genus has been proposed as resulting from the radiation of Neartic diaptomids which took place as post-Pliocene dispersal events involving repeated invasions of freshwater habitats in the region (*Suárez-Morales, 2003*; *Suárez-Morales & Reid, 2003*) and included Caribbean islands (*Bowman, 1986*). The latest dispersal of diaptomids in Middle America was likely to occur during the Holocene (8000 yr BP), probably an intermittent process linked to the alternative dry and wet periods and interglacial conditions. The genus *Mastigodiaptomus* is very well represented in the northern Neotropical region and there is a growing set of evidence of its radiation in Mexico and Central America (*Gutiérrez-Aguirre, Cervantes-Martínez & Elías-Gutiérrez, 2014*; *Gutiérrez-Aguirre & Cervantes-Martínez, 2016*; *Mercado-Salas et al., 2018*; *Gutiérrez-Aguirre et al., 2020*).

The emergence of the Galápagos islands resulted from volcanic processes that occurred in the eastern Pacific within the last 3–4 MY, during the Pleistocene. San Cristóbal island is one of the two oldest in the archipelago (emerged *ca.* 4 MYA). The earliest human activities associated to the Galápagos islands are related to Pre-colonization commerce (*Bush et al., 2022*).

According to *Lévêque, Bowman & Billeb (1966)* and *Wiedenfield & Jiménez-Uzcátegui (2008)*, the Galápagos archipelago harbors a diverse array of endemic and migrant birds (27–63 species), the highest among all other eastern Pacific oceanic islands. According to *Hessen, Jensen & Walseng (2019)*, the highly dynamic population sizes of waterfowl represent a relevant vector for zoochory of freshwater zooplankters, which in turn is useful to assess local and regional biogeographic patterns of zooplankton richness and community composition; this is especially true in isolated freshwater systems, like El Junco. Migrating birds can be a highly efficient factor for transporting zooplankton species, including copepods. Birds have higher potential for dispersal of zooplankters in drought conditions (*Morais-Junior et al., 2019*). Diaptomid diapausing eggs favors survival in harsh environmental conditions (*De Stasio, 1989*), including desiccation and transportation. However, how *M. galapagoensis* colonized El Junco remains an open question. The southernmost records of members of *Mastigodiaptomus* are from Guatemala, Central

America, where *M. amatitlanensis* seems to be extinct. Hence, no species of the genus have been known to occur south of Central America or in South American countries along the Eastern Pacific coast (*i.e.,* Colombia, Ecuador, Peru), so we cannot conclude whether *M. galapagoensis* was transported by zoochory from adjacent coastal areas or incidentally introduced by early sailors if it existed previously in the continent.

## CONCLUSIONS

Until now, no endemic freshwater zooplankters besides *M. galapagoensis* have been found in the archipelago, but all freshwater ecosystems (temporary or permanent) have not been thoroughly studied, particularly for zooplankton composition. So far, we know only the report on rotifers of *Segers (1991)*, a bachelor thesis *Obando (2009)*, and the checklists of cladocerans by *Lopez et al. (2018a)*, rotifers by *López et al. (2020)* and copepods (*Peck, 1994*; *Corgosinho et al., 2019*). We consider doubtful the records of marine copepods (*Acrocalanus* sp. and *Clausocalanus* sp.) from freshwater temporary ponds, not connected with the sea in San Cristóbal island (*Peck, 1994*).

Because of the biological relevance of this archipelago, we consider a priority to start intensive studies of the zooplankton (in a wide sense, and using modern collecting methods, see *Montes-Ortiz & Elías-Gutiérrez, 2018* and *Elías-Gutiérrez et al., 2018*) in all the archipelago, including the populated islands (*i.e.,* Santa Cruz, San Cristóbal, Isabela and Floreana) which are vulnerable despite governmental protective measures. An example was the illegal introduction of tilapia in the recent past, and the harmful method used for its eradication, its effects being documented only in a bachelor thesis (*Obando, 2009*) and a short paragraph of a book describing eradications of invasive species, like Tilapia, in Pacific islands (*Nico & Walsh, 2011*).

## ACKNOWLEDGEMENTS

The Charles Darwin Research Station and the Galápagos National Park provided logistic support to study El Junco Lake (Project PC 08-04). Humberto Bahena (ECOSUR-Chetumal) performed the final edition of the SEM plates.

### Funding

The biological material was obtained after funding from the Project Number PC 08-04 was granted to Miriam Steinitz-Kannan to study El Junco Lake. The visit of Manuel Elías-Gutiérrez to Ecuador, from which this paper emerged, was possible from funding from the project ''Biodiversidad invisible de aguas interiores de las islas Galápagos: Taxonomía integrativa y ecología'' proyecto CADS-1-2022 del Decanato de Investigación de ESPOL. The funders had no role in study design, data collection and analysis, decision to publish, or preparation of the manuscript.

## Grant Disclosures

The following grant information was disclosed by the authors:

El Junco Lake: PC 08-04.

Biodiversidad invisible de aguas interiores de las islas Galápagos: Taxonomía integrativa y ecología: CADS-1-2022 del Decanato de Investigación de ESPOL.

## Competing Interests

The authors declare there are no competing interests.

## Author Contributions

- Manuel Elías-Gutiérrez conceived and designed the experiments, performed the experiments, analyzed the data, prepared figures and/or tables, authored or reviewed drafts of the article, and approved the final draft.
- Miriam Steinitz-Kannan conceived and designed the experiments, analyzed the data, authored or reviewed drafts of the article, and approved the final draft.
- Eduardo Suárez-Morales conceived and designed the experiments, performed the experiments, analyzed the data, prepared figures and/or tables, authored or reviewed drafts of the article, and approved the final draft.
- Carlos López conceived and designed the experiments, analyzed the data, authored or reviewed drafts of the article, and approved the final draft.

## Field Study Permissions

The following information was supplied relating to field study approvals (i.e., approving body and any reference numbers):

Galápagos National Park approved the study (Project number PC 08-04).

## Data Availability

The specimens examined were deposited in the Collection of Zooplankton (ECO-CHZ) held at El Colegio de la Frontera Sur (ECOSUR), Chetumal, Mexico. A subsample from the same collection with numerous specimens is also deposited in the same collection (ECO-CHZ).

ECO-CH-Z 11820, ECO-CHZ-11821, ECO-CH-Z-11822, ECO-CH-Z-11823, ECO-CH-Z-11823, ECO-CH-Z-11824

## New Species Registration

The following information was supplied regarding the registration of a newly described species:

Publication LSID: urn:lsid:zoobank.org:pub:1889D5E7-1498-4F91-9782-9CA0877035B0

Species LSID: urn:lsid:zoobank.org:act:743005F4-CE38-4F41-843D-4B525E7E4546

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
