# Peer review of "Mastigodiaptomus galapagoensis n. sp. (Crustacea: Copepoda: Diaptomidae), a possibly extinct copepod from a crater lake of the Galápagos archipelago"

_PeerJ, doi:10.7717/peerj.15807_

## Round 0.1 · original submission · Major Revisions

The reviewers have done a thorough job and I feel that all of their concerns should be addressed (point by point). In particular, please pay attention to the required improvements of the drawings and suggested clarifications of the nomenclature. The suggested language editing should also be performed.

Reviewer 1 ·

Basic reporting

The theme of this study is quite interesting and valuable because of a description of limited specimens of a possibly extinct species from Galapagos islands. However, there are very many problems in the Description section. It is strongly recommended that all illustrations are reexamined by referring the corresponding figures of related congeners in previous studies. The problems are pointed out in Specific comments, by which I hope the manuscript will be improved.

Problems throughout the manuscript are as follows:
1) The segment shapes, such as subrectangular and subtriangular, were often used in the Description. I think these descriptions are ambiguous (not definitive) and do not have significance as a distinctive character of the species.
2) The terminology should be unified throughout the manuscript, e.g., exopod vs. exopodite and fifth leg vs. P5.
3) Characteristic features of some morphologies noted in the Discussion section are not described in the Description section, for example, spinous process length of the antepenultimate segment of male right antennule and length of aculeus on the male right P5. Morphologies cited in the Discussion must be described beforehand in the Description section.

Specific comments
Line 13: “11” should read “1”.
Line 117: “Normanski” may be misspelled “Nomarski”.
Line 144: A period should be added after “-Kannan”.
Line 144-145: Replace “Allotype. Adult female,” by “Allotype, adult female,”.
Line 163: “± 172” is a standard deviation or a standard error? Anyhow, I recommend to describe the length of the allotype and the range of other specimens rather than the mean length ± SD (or SE), because the most important information about the body length is of the holotype and the next is the allotype.
Line 169: The elements on the fifth pediger were described “long sensillum” and “small spinule” on the left wing, while on the right wing “two spines”. The difference between spinule and spine is their size, but sensillum has different functions. How do you distinguish them? They all look like small spines to me.
Line 172-173: The left spine on the genital double somite illustrated in Fig. 2M is larger than the right spine, which is contradictory with the description. In Fig. 3J, which is a ventral view, the right spine on the genital double somite is perpendicular to the body axis and the left one is directed posteriorly. Therefore, Fig. 2M should be a ventral view despite the caption noted “dorsal view”. Which is right?
Line 175: Left caudal ramus looks more narrower thar the right one. If so, please note it. Otherwise, please note that “rami symmetrical.”
Line 177: Caudal rami have 6 (not 5) caudal setae; dorsal one should be a caudal seta. Besides, the distance between the outermost proximal seta and the next one in Fig 2M, is significantly different between two rami. If so, it is very characteristic for the species. If not, Fig. 2M should be redrawn.
Line 180: According to Fig. 2A, the second segment looks having 1s+2ae instead of 2s+1ae. Besides, insert a comma after the segments 3 and 24, and replace “25(4s + 1ae)” by “25(4s, 1ae)”
Line 185: Delete “(Fig. 1F)”, which is absent.
Line 186: Please unify the terminology. Replace “Endopodal ramus” by “Endopod”.
Line 187: How did you recognize the 2nd segment as a fused segment? According to the SEM photo (Fig, 3F), the segment looks to be separated in two segments.
Line 187: Setal formula should read “1,2,1,1,1,1,1,4”, according to Fig. 3F.
Line 189: Coxa of the mandible is the segment with gnathobase in Fig. 2E, and basis is the first segment of palp. In Fig. 2D, delete the line between the 1st and 2nd segments, which seems to be incorrectly drawn. Endopod has 2 segments with 4, 5 setae, respectively, according to Fig. 2D. The setal formula of exopod 1, 1, 2, 2 is doubtful, because the general formula for diaptomids is 1, 1, 1, 3. Please reexamine the specimen.
Line 196: Please show a segmented line between two endopodal segments of the basis in Fig. 2F.
Line 198: Segmentation of maxilla in Fig. 2G is doubtful. The general endite pattern in diaptomids is 2 praecoxal, 2 coxal, 1 (or 2) basal endites. Please reexamine the specimen.
Line 202: The general setal formula on syncoxa of maxilliped is 1, 2, 3. 4.
Line 204: Endopod of maxilliped is 4 (not 5) segmented in Fig. 2C. Please show a segmented line between the 1st and 2nd segments.
Line 207: “exopodal pilose” should read “exopodal segments pilose”.
Line 208: I have not commented above, but I dare to say that morphologies common to the genus are not always necessary to describe a species if these are clearly shown in the figures. For example, segmentation of legs is common in the genus.
Line 217: No such citation should be made in an original description of a species. In fact, there are several differences between your figures and those in the cited paper.
Line 220: Please delete “coxal”, and replace “Basipod” by “Basis” to unify the terminology.
Line 222: Brush-like elements of the the 2nd endopodal segment are not “setae”. You might consider to use “elements” instead. If Fig. 2L is true, it is very important to describe the difference in the endopod length between the left and right legs, which are greatly longer in the right one. Please reexamine female P5 of the other specimens.
Line 222-225: Exopod in Fig. 2L is apparently wrong. The terminal claw is not a spine but an attenuation of the segment (see other diaptomid species). In addition, a short proximalmost seta along the lateral margin is forgotten. The third segment in Fig. 2L looks completely reduced into 2 setae. The 3rd segment is usually present as described in other congeners, but Fig. 2L shows only setae without recognizable segment. If the 3rd segment is completely reduced into 2 setae, the description should be like “exopod 2-segmented with 3 setae on 2nd segment; unequal distal two of them remnants of degenerated third segment.” Please reexamine female P5 and revise the figure.
Line 226: Please see the comment on Line 163.
Line 232: There are some differences between the description as Fig. 2O (segments 1,9,12,13,16,17,19,20,21) and the format of elements in parentheses (segments 1,11, 14,15,17) is not uniformed. Please reexamine the specimen to coincide the description and illustration. Besides, the figure should be prepared to recognize each element.
Line 235: If spinous processes are given in the seta formula of the previous sentence, then this sentence should be deleted.
Line 236: What is “peak-like”? Perhaps the beak-like is wrong?
Line 238: Two setae are seen on coxa in Fig. 2N. Please describe both.
Line 240: Not “setae” but “elements”. “4F” is probably mistaken for “4G”. Please add an enlarged figure of the second endopodal segment, because the segment is important for comparison with closely allied species in the Discussion section.
Line 243: “4H-K” should read “4H-J”.
Line 245: “4H-K” should read “4H”, because 4I-J are not on aculeus but on terminal claw. By the way, a line is seen near the base of aculeus and the spine bents at the line. Is this a morphology or a trace of broken? Anyway, please describe it after examining other specimens.
Line 246: “Basipod” should be “Basis”.
Line 248: “4G” should be “2N”.
Line 248-251: Please add a figure to show the complicate morphology of endopod.
Line 257: “left fifth exopodite” should be “left fifth leg”.
Line 260-261: Which segment of “left male antennule”?
Line 301-302: The sentence “This process is known to be…” should cite some references showing this.
Line 367: The incidental introduction by early sailors seems impossible. If the new species diverged from an ancestral species introduced from the Nearctic continent by early sailors, then thousand years is far too short a time to evolve into a new species.

Fig. 1B: What is the line like “2” or “z” on the second urosomites?
Figs. 1A & 2M: Length of the genital-double somite differs greatly between Figs. 1B and 2M. If this difference is observed in the other specimen, please describe it as a diagnostic feature. Otherwise, explain the reason of difference.
Fig. 2N caption: Revise to “Male fifth leg, anterior view”.

Experimental design

no comment

Validity of the findings

no comment

Additional comments

no comment

Reviewer 2 ·

Basic reporting

The English needs improvement. The authors should request editing help from a fluent speaker, to catch bothersome errors such as, on line 151, “designated”, not “designed”; line 117, “Nomarski” not “Nomanski’; lines 270-271, “southern” and “northern”, “Nearctic” not “Nearctics”; line 397, “Forty”, not “Fourty”; in several places, “on”, not “in” an island; etc.
Punctuation is inconsistent throughout but especially in the reference list. Other problems with references:
LL 75, 374, 411 – Corgosinho, not Corgoshinho
L. 305 - Gutiérrez-Aguirre et al., Fig. 11F – year??
LL. 359-60, 415 – de Morais should be cited as Morais-Junior S, and alphabetized under M
L 361, 419 – De Stasio, not Destasio
LL 421, 466 - Cite Edinaldo Nelson dos Santos-Silva as he customarily cites himself, as Santos-Silva EN (the “dos” is not necessary and is not capitalized)
LL 313, 324, 327 and following – There are 2 references “López et al., 2013”. Which is this? Designate “a” and “b”.
L 510 – Please correct the authorship -- the section on Calanoida was written by Mildred S. Wilson alone. (Harry Yeatman coauthored the entire chapter on Copepoda.) Correct the title to: Free-living Copepoda: Calanoida. Correct the spelling of Edmondson.

The references are well chosen and adequate. Surprisingly in a manuscript submitted by experienced researchers, several articles cited in the text are absent from the list of references:
LL. 297-8 - Santos-Silva et al., 1996
L. 352 - Lévequé et al. (1966)
L. 352 - Wiedenfeld (2006)
L. 373 - Segers (1991)
LL. 378-9 – Montes-Ortíz (2018); do you mean Montes-Ortíz and Elias-Gutiérrez (2018)?
L. 379 - Elias-Gutiérrez et al., 2019.

In general, the description is thorough. The lack of genetic information is certainly understandable. The comparison with similar congeners is clear, as is the justification for considering this as a separate and new taxon. However, I do not consider SEM photomicrographs to be an adequate replacement for clear line drawings; one reason is that specimens in SEMs are often oriented oddly, making it difficult to interpret structures; and another is that SEM preparation methods often distort structures. Here, some of the drawings could be substantially improved, especially with enlargements of critical structures:
Fig. 1A – is the last urosomite of the male really completely divided?
Fig. 2A, O – the drawings of the A1s are much, much too small. The relative sizes/lengths of the spines, presence/absence of setae, and processes are critical characters for distinguishing among diaptomid species, and are discussed in detail in the text – but are nearly invisible in the drawings. The SEMs do not substitute for this lack. Much bigger, please!
Fig. 2C-G – arrange the mouthparts anteriorly-posteriorly as they are on the animal.
Fig. 2H-K – are these anterior or posterior views of legs 1-4? Is Schmeil’s organ present on P2?
Fig. 2L - P5 of the female: the endopodites appear to be of different lengths, possibly a function of different angles of view. Please clarify this in the text. Also, please provide a detailed view of the ornamentation on the end of the endopodite.
Fig. 2M – the dorsal setae on the caudal rami are nearly invisible. Please enlarge.
Fig. 2N - P5 R exp – Both anterior and posterior views are needed. R P5 - is the terminal claw really segmented near the base? Fine detail of the ornamentation of the R P5 endopodite and L P5 exo- and endopodites must be illustrated (enlarged). Are the medial surfaces of the bases devoid of fine ornamentation?

Experimental design

No comment

Validity of the findings

The story is well told, with a well written introduction, thorough description of the habitat, and appropriate detail for the specialist. The discussion of transport possibilities usefully summarizes experimental work and inferences from the literature.

Additional comments

This description of a species of diaptomid copepod would be of interest only to specialists were it not for the unusual circumstances of the apparent elimination of its habitat and the fortuitous preservation of the specimens. As such, this manuscript provides an object lesson for those concerned with habitat preservation and specifically to everyone who makes and stores biological collections.

---

## Round 0.2 · Minor Revisions

Thank you for the corrections and additions to the original manuscript and the consideration of the reviewers' suggestions. This has indeed resulted in an improved version of the manuscript.

Nevertheless, there seem to be some minor issues that should be addressed. As recommended by the reviewer in the second round of the review process, please correct the remaining issues, especially the following:

(1) check the orientation of the urosoma in Fig. 2M (dorsal/ventral?)

(2) in Fig. 2G (instead of Fig. 2C) check the segmentation of the maxillipedal terminal segment

(5) delete the ambiguous z (or 2?) in the middle of the second urosomite in Fig. 1B, as it is obviously confusing and not mentioned in the description. I was also quite curious about this feature...

(6) add the reference to the curved urosoma in parentheses to the legend of Fig. 1B

Line 197: delete "of the latter"

Line 199: move citation of Fig. 2D to the end of previous sentence.

The text also contains some minor proofreading-level errors (the line numbers correspond to those in the revised version in pdf format:

L 150 - do not capitalize the word adult in "Allotype, Adult female"

L 329 - indicate either "a" or "b" for the reference to López et al, 2018

LL 186-188 and LL237-239 (A1 armature) - check for an extra space between the segment number and the bracket (e.g., 1(1s, 1ae), 2(2s, 1ae), 3 (2s, 1ae), 4-5(1s).... Alternatively, you can insert a space everywhere.

L 224 - delete the word Fig., before Female P5; also correct the typo in "symmetrical"

L230 - "setae" should be changed to singular "seta" (one long and one short seta)

L242 - delete two punctuation marks here: . . Left A1, mouthparts, and P1-P4 as in female.

L251 - do not capitalize the word left in "Male Left P5"

L459 - presumably López (not Lopez)

I hope to see the re-revised version soon.

Reviewer 1 ·

Basic reporting

no comment

Experimental design

no comment

Validity of the findings

no comment

Additional comments

Comments for revised MS were in the attached pdf.

Annotated reviews are not available for download in order to protect the identity of reviewers who chose to remain anonymous.

---

## Round 0.3 · accepted · Accept

Many thanks for your detailed work and attention to the suggested revisions. Your manuscript is now ready for publication.